# SPEED: Sharpened-Teacher Distillation for Parallel Decoding of Diffusion Language Models

Qiuhong Shen [1]   Xingyi Yang [2]   Xinyin Ma [1]   Gongfan Fang [1]   Xinchao Wang [1]

## Abstract

Diffusion-based large language models generate text by gradually filling in masked tokens, yet they remain slow because they usually decode only a few tokens per step. Parallel decoding, which unmasks multiple tokens simultaneously, promises acceleration but often degrades quality when too many tokens are predicted at once. We identify the root cause: when decoding is viewed as iterative token grouping, overly permissive grouping places interdependent tokens in the same step, violates the conditional independence assumption, and amplifies reliance on noisy context even when the top prediction is already correct. We introduce *SPEED*, a framework that enlarges safe parallel groups through complementary training and inference designs. At training time, a sharpened teacher distillation objective selectively aligns the student to teacher-correct positions using a temperature-scaled KL term together with a masked language modeling loss, producing a student that assigns more probability mass to correct token identities and elevates more positions above the decoding threshold. At inference time, *Slow–Fast Decoding* partitions tokens by sensitivity to revealed context using token-wise Jensen–Shannon Divergence computed with and without access to the preceding block, decoding low-sensitivity tokens jointly in parallel while deferring high-sensitivity tokens until sufficient context resolves them. Through extensive experiments, our framework attains up to 12.2× speedup on LLaDA-8B-Instruct and 6.7× on Dream-7B-Instruct with accuracy close to greedy decoding across standard reasoning and code benchmarks. Code and results are available at https://github.com/florinshen/SPEED.

[1]National University of Singapore [2]The Hong Kong Polytechnic University. Correspondence to: Xinchao Wang <xinchao@nus.edu.sg>.

*Proceedings of the 43rd International Conference on Machine Learning*, Seoul, South Korea. PMLR 306, 2026. Copyright 2026 by the author(s).

## 1. Introduction

Generative models for natural language have become the cornerstone of modern artificial intelligence, enabling a vast array of applications. Among these, Masked Diffusion Models (MDMs) (Nie et al., 2025b; Ye et al., 2025; Gulrajani & Hashimoto, 2023; He et al., 2023; Li et al., 2025b) have emerged as a powerful and promising paradigm. By iteratively denoising a sequence from a fully masked state, MDMs offer a highly parallelizable framework for generation. This inherent parallelism presents a significant advantage, holding the potential for substantial improvements in generation speed and efficiency, a critical factor for the deployment of large-scale language models in real-world scenarios.

In practice, however, this potential for speed remains underutilized because most MDMs decode only a few tokens at each step. Typically, the sequence is divided into blocks, and within each block, tokens are revealed incrementally over multiple steps. Confidence-aware parallel decoding (Wu et al., 2025; Yu et al., 2025) accelerates this by unmasking all tokens whose predicted probability exceeds a high threshold. Yet, pushing for greater speed, by lowering the threshold to decode more tokens per step, invariably leads to a sharp drop in generation quality. This sharp speed–quality trade-off suggests that current models are not truly ready for aggressive parallelism.

We identify the root cause: dependencies between tokens decoded in the same step. To analyze this, we introduce the perspective of viewing MDM decoding as an *iterative token grouping* process, where the goal at each step is to identify the largest possible group of tokens that can be predicted in parallel without sacrificing coherence. The performance degradation at low confidence thresholds occurs when this grouping is suboptimal, forcing tokens with strong inter-dependencies to be decoded simultaneously, thereby violating the underlying conditional independence assumption. Our central hypothesis is that standard training objectives promote correlated reliance on noisy context, which leaves tokens uncertain even when the argmax token identity is already correct.

To address this, we propose SPEED with Jensen–Shannon

Divergence JSD based decoding, a unified framework that improves training and inference to enable safe parallelization. Our goal is to increase the size of token groups decoded per step while preserving coherence and generation quality.

In training, we introduce sharpened teacher distillation. A frozen teacher produces targets that are selectively distilled to the student using a temperature-scaled KL objective applied only to the subset tokens where the teacher is already predicted correctly. This selective supervision concentrates learning on tokens that can be made confidently predictable, while a masked language modeling term on the active block maintains faithfulness to ground truth.

At inference, we complement training with a Slow–Fast Decoding strategy that partitions tokens by their sensitivity to the already generated context. We measure sensitivity using JSD between the predictive distributions with and without access to the preceding block. Tokens with low JSD are stable across contexts and are decoded jointly in parallel, while tokens with high JSD are context-sensitive linchpins whose distributions are likely to shift further as peer tokens resolve, so they are deferred and decoded more cautiously. This adaptive grouping admits larger parallel steps where safe, improving throughput while preserving accuracy.

Our contributions are threefold and can be summarized as follows:

- We introduce a perspective that frames MDM decoding as an *iterative token grouping* problem, and identify suboptimal grouping at permissive thresholds as the core bottleneck that violates conditional independence.

- We propose sharpened teacher distillation, which applies a temperature-scaled KL objective selectively on the teacher-correct subset to guide the student to place more probability mass on correct token identities and thereby enlarging safely decodable groups.

- We develop a Slow-Fast Decoding, an inference strategy that uses Jensen–Shannon Divergence to group only those tokens that are truly ready for parallel decoding, preserving quality while accelerating generation.

- Through extensive experiments, we show that the combined approach advances the speed–quality frontier for MDMs, achieving up to $12.2\times$ acceleration on MBPP with accuracy within 1.2 points of greedy and $9.8\times$ acceleration on GSM8K with a 1.4-point difference, outperforming existing baselines.

## 2. Related works

**Discrete Diffusion Language Models.** Discrete diffusion languguage models (dLM) (Devlin et al., 2019; Ye et al., 2025; Nie et al., 2025b; Liu et al., 2025a; Arriola et al., 2025; Chen et al., 2026b; Yu et al., 2026) have recently been a compelling paradigm for non-autoregressive text generation. Unlike previous left-to-right generation in autoregressive models (Radford et al., 2019; Yang et al., 2025; Leviathan et al., 2023; Yan et al., 2025), these dLM models operate by iteratively refining a sequence from a corrupted state, typically one filled with [MASK] tokens. Pioneering works (Lovelace et al., 2023; 2024; Gong et al., 2025; Nie et al., 2025a; Chen et al., 2026a; Wang et al., 2026) established the scalability of the masked diffusion language models, demonstrating that these models could effectively leverage large-scale data and parameter counts. With the demonstrated scalability, a series of new powerful diffusion language models (Nie et al., 2025b; Ye et al., 2025; Song et al., 2025; Khanna et al., 2025; Zhu et al., 2025) have emerged. Most notably, recent open-source large diffusion language models such as LLaDA (Nie et al., 2025b) and Dream (Ye et al., 2025) have achieved performance that is highly competitive with autoregressive counterparts of comparable model scales, underscoring their viability as a promising architecture for generative language tasks. Our work builds upon these works, addressing the critical challenge of inference latency that currently limits their practical deployment.

**Acceleration of Masked Diffusion Models.** Despite their strong performance, a primary challenge for Masked Diffusion Models is their inference latency, which often trails that of highly optimized autoregressive models. This latency stems from two factors. First, the non-autoregressive nature of the decoding process precludes the use of standard KV-caching mechanisms. Several works have proposed specialized caching variants to reduce redundant computations in this setting (Ma et al., 2025; Liu et al., 2025b; Wu et al., 2025; Wang et al., 2025). Second, and more central to our work, is the bottleneck within the iterative decoding process itself. Previous approaches (Nie et al., 2025b; Ye et al., 2025) often employ a greedy decoding strategy, decoding only the single most confident token per step, which is computationally inefficient. Confidence-aware parallel decoding (Wu et al., 2025) mitigates this by simultaneously unmasking all tokens whose predicted confidence exceeds a high threshold. However, this approach is constrained by a sharp speed-performance trade-off: lowering the threshold to increase parallelism and accelerate inference invariably leads to a significant degradation in generation quality. Our work explores to tackle this challenge to enabling it to confidently generate larger groups of tokens per step by reshaping its learned tokens dependencies through distillation and grouping constrain.

# 3. Masked Diffusion Models Decoding as Iterative Token Grouping

Masked Diffusion Models (MDMs) (Nie et al., 2025b; Ye et al., 2025) have recently emerged as a powerful class of generative models for natural language, demonstrating compelling performance on a diverse range of tasks. MDMs operate via a forward noising process that incrementally corrupts an input sequence $\boldsymbol{x}_0$ by replacing its tokens with a special [MASK] token. This process is governed by a predefined noise schedule, and the distribution of a noisy sequence $\boldsymbol{x}_t$ at time $t \in [0, 1]$ conditioned on the original sequence $\boldsymbol{x}_0$ is given by:

$$
\begin{aligned}
q(\boldsymbol{x}_t \mid \boldsymbol{x}_0) &= \prod_{i=1}^{n} q(\boldsymbol{x}_t^i \mid \boldsymbol{x}_0^i) \\
&= \prod_{i=1}^{n} \mathrm{Cat}\Big(\boldsymbol{x}_t^i; (1-t)\delta_{\boldsymbol{x}_0^i} + t\,\delta_{[\mathrm{MASK}]}\Big)
\end{aligned} \quad (1)
$$

Here, $t$ represents the continuous diffusion time (or noise level), controlling the interpolation between the clean data distribution at $t = 0$ and a fully masked sequence at $t = 1$.

The reverse process, which generates a clean sequence from a fully masked input $\boldsymbol{x}_1$, is learned by a model $p_\theta$. Decoding is typically performed in a semi-autoregressive manner. The sequence is partitioned into $N$ contiguous blocks, $\{B_1, \ldots, B_N\}$. These blocks are generated sequentially. Within each block $B_i$, the masked tokens are denoised over multiple steps. The generation of block $B_i$ is conditioned on the previously generated blocks $\{B_1, \ldots, B_{i-1}\}$ and the still-masked future blocks $\{B_{i+1}, \ldots, B_N\}$:

$$
p_\theta(\boldsymbol{x}_{B_i} \mid \boldsymbol{x}_{B_{<i}}, \boldsymbol{x}_{B_{>i}}^{\mathrm{masked}}) = \prod_{k=1}^{M_i} p_\theta(\boldsymbol{x}_{t_{k-1}, B_i} \mid \mathrm{context}), \quad (2)
$$

where $\mathrm{context} = \{\boldsymbol{x}_{t_k, B_i}, \boldsymbol{x}_{B_{<i}}, \boldsymbol{x}_{B_{>i}}^{\mathrm{masked}}\}$, $\boldsymbol{x}_{B_{<i}}$ denotes the set of fully denoised preceding blocks, $\boldsymbol{x}_{B_{>i}}^{\mathrm{masked}}$ denotes the subsequent masked blocks, $1 = t_{M_i} > \cdots > t_1 > t_0 = 0$ is a discrete reverse timestep schedule, and $M_i$ is the number of denoising steps for block $B_i$. A standard greedy approach reveals one token with the highest model confidence at each step, making the number of steps equal to the block length ($M_i = |B_i|$). This sequential intra-block decoding is a significant computational bottleneck.

To mitigate this, confidence-aware parallel decoding strategies (Wu et al., 2025; Yu et al., 2025) have been proposed. At each step, all masked tokens with a predicted probability exceeding a certain threshold $\tau$ are decoded simultaneously. If no token's confidence surpasses $\tau$, only the single most confident token is decoded. As theoretically justified (Wu et al., 2025), for a high threshold $\tau = 1 - \epsilon$, the predictions for selected tokens are approximately conditionally

independent. This allows for parallel decoding that closely approximates the greedy sequential process, achieving significant speedups (e.g., $3\times$) with negligible performance degradation for high $\tau$ (e.g., $\tau = 0.9$).

This confidence-aware decoding implicitly performs a dynamic token grouping. The key to accelerating MDM decoding lies in minimizing the number of sequential decoding steps, $M_i$, for each block. This is equivalent to finding an optimal partition of the tokens within a block. Let the set of token indices in block $B_i$ be $\mathcal{I}_i$. The decoding process partitions $\mathcal{I}_i$ into an ordered sequence of disjoint groups $\mathcal{P}_i = (G_1, G_2, \ldots, G_{M_i})$, where $\mathcal{I}_i = \bigcup_{k=1}^{M_i} G_k$. The generation of the block can then be expressed as:

$$
p_\theta(\boldsymbol{x}_{B_i} \mid \mathrm{context}) = \prod_{k=1}^{M_i} p_\theta(\boldsymbol{x}_{G_k} \mid \boldsymbol{x}_{G_{<k}}, \mathrm{context}), \quad (3)
$$

where $\boldsymbol{x}_{G_k}$ are the tokens corresponding to indices in group $G_k$, and $\boldsymbol{x}_{G_{<k}}$ are all previously decoded tokens in the block. The parallel decoding strategy makes a crucial conditional independence assumption within each group:

$$
p_\theta(\boldsymbol{x}_{G_k} \mid \boldsymbol{x}_{G_{<k}}, \mathrm{context}) \approx \prod_{j \in G_k} p_\theta(x_j \mid \boldsymbol{x}_{G_{<k}}, \mathrm{context}).
$$
$$ (4) $$

The number of sequential steps is thus $M_i = |\mathcal{P}_i|$, the number of groups in the partition.

However, a fundamental tension exists. Directly lowering the confidence threshold $\tau$ reduces $M_i$ by creating larger, more inclusive groups, but it often leads to a sharp decline in generation quality. This performance drop occurs because a lower threshold is more likely to group tokens with strong inter-dependencies into the same step $G_k$. This violates the independence assumption in Eq. 4, causing the model to generate inconsistent or incoherent text. We argue that fewer decoding steps can be achieved by explicitly encouraging the model to assign higher confidence to tokens that are jointly decodable. This intuition aligns with observations in prior work (Li et al., 2025a), which shows that many tokens predicted with low confidence in early steps are correct. By learning to recognize and promote such tokens, the model can effectively reduce $M_i$ without compromising output quality. Figure 1 provides empirical evidence that this is achievable: our proposed framework SPEED (detailed in Sec. 4.1) preserves higher self-consistency at larger parallel group sizes and elevates substantially more tokens above the decoding threshold than the base model.

# 4. Methodology

To achieve fewer sequential decoding steps in parallel decoding, we propose solutions at both training and inference time.

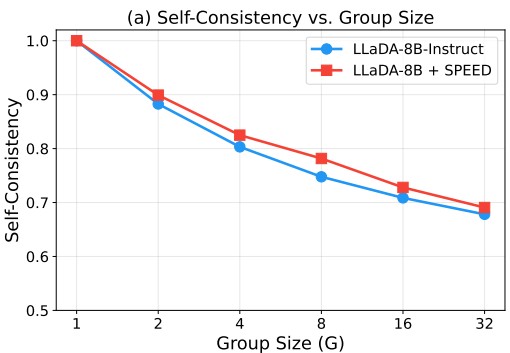 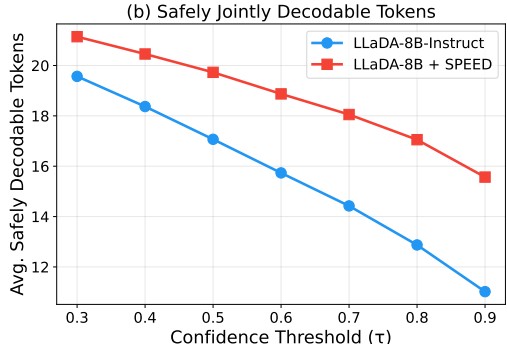

*Figure 1.* **Effect of SPEED on safe parallel decoding.** (a) Self-consistency between the joint parallel prediction and the sequential greedy prediction as a function of parallel group size $G$. The baseline degrades quickly as $G$ grows, whereas SPEED preserves higher self-consistency at every $G$. (b) Average number of tokens in a block whose confidence already exceeds the decoding threshold $\tau$. SPEED elevates 2–4 additional tokens above every threshold value, directly enlarging the safe parallel group and reducing $M_i$. See Appendix A.2 for the construction of both diagnostics and the underlying numerical values.

**Training.** First, we distill a student model from a progressively sharpened teacher (Sec. 4.1) so that the student learns to concentrate probability mass on tokens that the teacher already predicts correctly. Concretely, a KLD loss focused on teacher-correct positions ($\mathcal{M}_c$) sharpens confidences, while a masked language modeling cross-entropy term preserves overall correctness. The combined objective encourages the student to raise token probabilities above the decoding threshold $\tau$ without sacrificing accuracy, thereby increasing the expected parallel group size and reducing $M_i$.

**Inference.** Second, we propose Slow-Fast Decoding in Sec 4.2, an inference strategy that dynamically groups tokens based on their sensitivity to already-generated context. As formalized in Eq. 3, we use Jensen–Shannon Divergence (JSD) to measure the dependency between a token's predictive distribution with and without access to the preceding block. Tokens with high JSD are context-stable and decoded in parallel ("fast"); those with low JSD are ambiguous and decoded sequentially ("slow"). This adaptive grouping ensures only compatible tokens are processed together, preserving generation quality while enabling acceleration.

### 4.1. Distillation from a Sharp Teacher

Our objective is to teach the model to assign high confidence to tokens that can be reliably decoded early during the semi-autoregressive denoising of a block. To formalize this, consider a group of tokens $G = \{g_1, \ldots, g_{|G|}\}$ within an active block. A cautious sequential teacher $\theta$ models the joint distribution over $G$ via a dependent factorization:

$$p_\theta(\boldsymbol{x}_G \mid \text{context}) = \prod_{j=1}^{|G|} p_\theta\big(x_{g_j} \mid \boldsymbol{x}_{\{g_1, \ldots, g_{j-1}\}}, \text{context}\big), \tag{5}$$

whereas an ideal parallel student $\theta^+$ would adopt a factorized, conditionally independent form:

$$p_{\theta^+}(\boldsymbol{x}_G \mid \text{context}) = \prod_{j=1}^{|G|} p_{\theta^+}\big(x_{g_j} \mid \text{context}\big). \tag{6}$$

Our aim is to train the student so that, under the factorized form in Eq. 6, each correctly predicted token in $G$ attains a sufficiently large predictive probability (above the decoding threshold $\tau$), thereby maximizing the expected group size decoded in parallel at each step.

To realize this objective we distill from a frozen teacher of identical architecture whose output distributions are progressively sharpened. We first construct an offline dataset of prompt–response trajectories produced by the base model under the semi-autoregressive decoding regime. For each generated target sequence $\boldsymbol{x}_0$ we sample a random block $B_i$, keep preceding blocks $B_{<i}$ intact, mask subsequent blocks $B_{>i}$, and apply noise at a random diffusion level $t \in [0, 1]$ to the active block $B_i$. The resulting input $\boldsymbol{x}_t$ and its masked index set for the active block $\mathcal{M}_t^a$ mimic the model's inference-time conditions.

During distillation the frozen teacher computes logits $\boldsymbol{z} = f_\theta(\boldsymbol{x}_t)$ in a single forward pass and we identify the subset of masked positions that the teacher already predicts correctly:

$$\mathcal{M}_c = \big\{i \in \mathcal{M}_t^a \mid \arg\max_v \boldsymbol{z}_v^i = x_0^i\big\}.$$

These indices correspond to tokens for which the teacher is both confident and accurate, and they are the primary targets for confidence sharpening. The student produces logits $\boldsymbol{z}^+ = f_{\theta^+}(\boldsymbol{x}_t)$ on the same input. To encourage the student to mirror the teacher's concentrated confidence on the correct subset, we minimize a temperature-scaled KL divergence between the teacher and student predictive distributions on $\mathcal{M}_c$:

$$\mathcal{L}_{\text{KLD}} = \frac{1}{|\mathcal{M}_c|} \sum_{k \in \mathcal{M}_c} \text{KL}\Big(\sigma\big(\boldsymbol{z}_k/T_t\big) \,\big\|\, \sigma\big(\boldsymbol{z}_k^+/T_s\big)\Big), \tag{7}$$

where $\sigma(\cdot)$ denotes the softmax operator, $T_t < 1$ is the teacher temperature used to sharpen the teacher distribution, and $T_s > 1$ is the student temperature used to produce a relatively flatter student distribution. During distillation

the teacher's parameters are periodically updated from the student's parameters, yielding progressively sharper teacher targets that guide the student toward more decisive, jointly-decodable predictions. By focusing the KL term on $\mathcal{M}_c$ we concentrate the sharpening signal on positions where the teacher is already correct, thereby reducing the risk of amplifying teacher mistakes.

Complementing the sharpening objective, we enforce correctness across the entire active block with a standard masked language modeling loss that directly supervises the student against the ground-truth tokens:

$$\mathcal{L}_{\text{MLM}} = -\frac{1}{|\mathcal{M}_t^a|} \sum_{k \in \mathcal{M}_t^a} \log \left[ \sigma(\boldsymbol{z}_k^+)_{x_0^k} \right]. \tag{8}$$

This cross-entropy term preserves fidelity to the data distribution and prevents the student from drifting toward spurious high-confidence predictions that do not match the ground truth.

The combined distillation objective is

$$\mathcal{L}_{\text{distill}} = \mathcal{L}_{\text{KLD}} + \beta \mathcal{L}_{\text{MLM}}, \tag{9}$$

with $\beta > 0$ balancing confidence sharpening and token correctness. Intuitively, the KL term teaches the student to concentrate probability mass on tokens that the teacher already decodes correctly (thereby raising those tokens above the decoding threshold $\tau$), while the MLM term preserves overall accuracy across the active block.

### 4.2. Slow-Fast Decoding based on JSD

While distillation from the sharpened teacher model effectively encourages the model to put more probability mass on correctly predicted token IDs, confidence scores or predictive entropy remain imperfect proxies for the conditional independence required by parallel decoding. In particular, at more permissive thresholds $\tau$ (e.g., lowering the confidence threshold or raising entropy threshold), tokens with strong but unresolved dependencies can be grouped together incorrectly, producing coherence failures and reducing generation quality. To mitigate this risk, we introduce a Jensen-Shannon divergence (JSD)–based constraint that more directly measures contextual dependency and thus guides safer token grouping.

Our key insight is that within any active block $B_i$, the uncertainty of a masked token stems from two distinct sources, namely the already-generated context from previous blocks ($\boldsymbol{x}_{B_{<i}}$) and the yet-to-be-resolved context from peer tokens inside the same block. A token whose distribution barely moves once the preceding block is revealed is largely shaped by global structure rather than local context, and it is therefore a safe candidate for parallel decoding. In contrast, a token that responds sharply when the preceding block

becomes available is a context-sensitive "linchpin" whose identity is likely to shift again as peer tokens in the same block are resolved, which makes it risky to commit early under the conditional independence assumption. To make this distinction operational, we quantify each token's contextual sensitivity using the Jensen-Shannon Divergence (JSD), which measures the gap between a token's predictive distribution with and without access to the denoised previous block.

Formally, for each masked token $j$ in the active block $B_i$, we compute its token-wise JSD as:

$$
\begin{aligned}
\mathcal{J}_j = \text{JSD}\Big( & p_\theta\big(\cdot \mid \boldsymbol{x}_{B_{<i}}, \boldsymbol{x}_{B_i \setminus \{j\}}^{\text{masked}}, \dots \big) \\
& \Big\| p_\theta\big(\cdot \mid \boldsymbol{x}_{B_{<i}}^{\text{masked}}, \boldsymbol{x}_{B_i \setminus \{j\}}^{\text{masked}}, \dots \big)\Big)
\end{aligned}
\tag{10}
$$

where $p_\theta(\cdot|\text{context})$ is the model's predicted probability distribution for token $j$. A high $\mathcal{J}_j$ indicates that the model's prediction for token $j$ changes significantly once the prior context $\boldsymbol{x}_{B_{<i}}$ is revealed, marking $j$ as highly dependent on cross-block context. Such tokens are linguistically pivotal, and their distributions are likewise expected to remain sensitive to additional intra-block evidence, which makes them unsafe to commit early inside a large parallel group. A low $\mathcal{J}_j$ instead suggests that the token is relatively stable and constrained primarily by the global structure of the sentence rather than the specific preceding words, which makes it a strong candidate for aggressive parallel decoding.

Building on this insight, we leverage the JSD metric to design a hybrid decoding strategy. Instead of applying a single threshold uniformly across the block, we partition the masked tokens in active block $B_i$ into a "fast set" $\mathcal{S}_{\text{fast}}$ of context-stable tokens and a "slow set" $\mathcal{S}_{\text{slow}}$ of context-sensitive ones, where the split is determined by a JSD partition threshold $\tau_{\mathcal{J}}$:

$$
\begin{aligned}
\mathcal{S}_{\text{fast}} &= \{j \in B_i \mid \mathcal{J}_j < \tau_{\mathcal{J}}\}, \\
\mathcal{S}_{\text{slow}} &= B_i \setminus \mathcal{S}_{\text{fast}}.
\end{aligned}
\tag{11}
$$

Tokens in the fast set carry low JSD and are robust to cross-block context, which suggests that their predictions will also remain stable as peer tokens in $B_i$ are resolved. We therefore decode them under an aggressive threshold $\tau_{\text{fast}}$, allowing wide parallel groups. Tokens in the slow set carry high JSD and are likely to shift further as additional intra-block evidence accumulates, so we process them under a conservative threshold $\tau_{\text{slow}}$ that defers their commitment to later steps. Once every fast token in $B_i$ has been committed, the remaining slow tokens are decoded with the threshold relaxed back to $\tau_{\text{fast}}$, because by that point their intra-block dependencies have been largely resolved. With this design, context-stable tokens accelerate the block while context-sensitive ones receive cautious, sequential treatment.

The effectiveness of this JSD-based grouping is not purely empirical. It is theoretically grounded in the goal of minimizing the error introduced by the conditional independence assumption in parallel decoding. We provide a formal justification in *Appendix* Sec. A.1.

## 5. Experiments

### 5.1. Experiments Setup

**Training Details.** Our experiments are conducted on two representative diffusion language models, LLaDA-8B-Instruct (Nie et al., 2025b) and Dream-7B-Instruct (Ye et al., 2025). For sharpened teacher distillation (SPEED), all sequences are padded to a fixed length of 512 and masked in a semi-autoregressive manner with a block size of 32. Within each active block, the masking ratio is fixed at 0.5. To minimize distribution shift during training, we adopt Low-Rank Adaptation (LoRA) with a rank of 128, a scaling factor of 128, and a dropout rate of 0.05. We use a temperature of $T_t = 0.1$ to sharpen the teacher distribution and $T_s = 1.5$ to flatten the student distribution in Eq. 7. The balance factor $\beta$ in the final loss (Eq. 9) is set to 1.5. We train for 5 epochs with a fixed learning rate of $2 \times 10^{-5}$. The teacher model is updated with the student weights at the end of each epoch to progressively refine the target distribution. Training is performed on 4 H200 GPUs and takes about 6 hours.

**Data Preparation.** For the prompt–response paired training data, we sample prompts from the GSM8K (Cobbe et al., 2021) training set, the math split of PRM800K (Lightman et al., 2023), and a subset of the Numina-Math (Li et al., 2024) dataset. We then use the teacher model itself to generate the corresponding responses. For generation configuration, we use semi-autoregressive decoding with a block size of 32 and apply threshold decoding for LLaDA-8B-Instruct (Nie et al., 2025b), while using greedy decoding for Dream-7B-Instruct (Ye et al., 2025). After generation, we filter out samples with incorrect final answers and retain only those with correct responses as effective training data. This results in approximately 120K training pairs for LLaDA and 90K for Dream.

**Inference Details.** During inference, we employ entropy-based thresholding with semi-autoregressive decoding on both distilled models. For LLaDA, we use a fixed entropy threshold of $\tau = 0.5$, while for Dream, we set $\tau = 0.45$. The JSD-based constraint introduces no additional forward passes at inference. At the first decoding step of each block $B_i$, the model produces logits not only for $B_i$ but also for the still-masked subsequent block $B_{i+1}$. We cache these distributions and reuse them at the first step of decoding $B_{i+1}$ to compute, for every masked token $j \in B_{i+1}$, the JSD between its distribution with $B_i$ masked (the cached logits) and its distribution with $B_i$ revealed (the fresh logits).

The dependency estimate is therefore obtained without any extra model evaluation. We then partition the tokens in $B_{i+1}$ into a fast set of low-JSD tokens and a slow set of high-JSD tokens using a fixed absolute cutoff $\tau_{\mathcal{J}} = 0.3 \ln 2$. The fixed cutoff exploits the bounded range of JSD on $[0, \ln 2]$ and avoids forcing a partition inside blocks whose tokens are tightly clustered in sensitivity. Fast-set tokens are decoded under the aggressive threshold $\tau_{\text{fast}} = \tau$ to enable wide parallel groups, while slow-set tokens are deferred under the conservative threshold $\tau_{\text{slow}} = \tau/2$. The first block has no preceding block to gate on, so we decode it uniformly under $\tau_{\text{fast}}$. Once every fast token in $B_{i+1}$ has been committed, the remaining slow tokens are decoded under $\tau_{\text{fast}}$ as well, because by that point their intra-block dependencies are largely resolved.

### 5.2. Main Results and Analysis

**Evaluation Benchmarks.** Following common evaluation protocols, we evaluate our method on four representative benchmarks spanning mathematical reasoning and code generation: GSM8K (Cobbe et al., 2021), HumanEval (Chen et al., 2021), MATH (Lewkowycz et al., 2022), and MBPP (Austin et al., 2021). These benchmarks are widely adopted to assess both the reasoning capability and generation accuracy of large language models.

**Comparison with Baselines.** We compare generation accuracy and efficiency against four parallel-decoding baselines: 1) Greedy decoding as used in the original works, which reveals a single token per step by selecting the token with maximum confidence or minimum entropy. 2) Few-step decoding, which directly shortens the number of generation steps to $1/4$ by revealing the top-4 tokens at each step. 3) Threshold-decoding (Wu et al., 2025), which applies semi-autoregressive generation with a fixed block size and decodes in parallel all tokens whose confidence exceeds a preset threshold for LLaDA or whose predictive entropy falls below a preset threshold for Dream. 4) Consistency decoding based on progressive self-distillation (Deschenaux & Gulcehre, 2025) along time. Following the setup in (Deschenaux & Gulcehre, 2025), we keep the LoRA configuration and training data consistent with our method and perform two rounds of distillation to reduce the decoding steps to $1/4$.

**Evaluation results on LLaDA.** Table 1 shows that our distilled LLaDA model achieves significant decoding acceleration while maintaining competitive accuracy across all four benchmarks. On GSM8K, greedy decoding requires 256 steps to reach 79.3% accuracy with a throughput of 1.5 tokens/s. In contrast, our method (SPEED) reduces the number of decoding steps to 26, achieving a $9.8\times$ speedup (16.1 tokens/s) with only a 1.4-point drop in accuracy (77.9%). It also substantially outperforms Few-step Decoding (65.6%)

| Benchmark | Method | #Steps↓ | Tokens/s↑ | Speedup↑ | Accuracy↑ |
|---|---|---|---|---|---|
| GSM8K (5-shot) | LLaDA-8B-Instruct | 256 | 1.5 | 1.0× | 79.3 |
| | Few-step Decoding | 64 | 5.8 | 4.0× | 65.6 |
| | Threshold Decoding | 77 | 4.9 | 3.3× | 78.7 |
| | Consistency Distill | 64 | 5.3 | 4.0× | 69.2 |
| | **SPEED (ours)** | **26** | **16.1** | **9.8×** | **77.9** |
| | **SPEED + JSD Decoding** | **29** | **14.3** | **8.8×** | **78.9** |
| MATH (4-shot) | LLaDA-8B-Instruct | 256 | 1.8 | 1.0× | 33.5 |
| | Few-step Decoding | 64 | 6.7 | 4.0× | 26.6 |
| | Threshold Decoding | 96 | 11.2 | 2.7× | 33.5 |
| | Consistency Distill | 64 | 6.5 | 4.0× | 27.2 |
| | **SPEED (ours)** | **36** | **17.9** | **7.1×** | **32.0** |
| | **SPEED + JSD Decoding** | **39** | **15.8** | **6.6×** | **32.8** |
| HumanEval (0-shot) | LLaDA-8B-Instruct | 256 | 5.2 | 1.0× | 42.6 |
| | Few-step Decoding | 64 | 18.9 | 4.0× | 18.3 |
| | Threshold Decoding | 77 | 17.1 | 3.3× | 42.6 |
| | Consistency Distill | 64 | 17.8 | 4.0× | 18.2 |
| | **SPEED (ours)** | **29** | **46.1** | **8.8×** | **40.9** |
| | **SPEED + JSD Decoding** | **31** | **43.0** | **8.3×** | **42.1** |
| MBPP (3-shot) | LLaDA-8B-Instruct | 256 | 2.1 | 1.0× | 42.4 |
| | Few-step Decoding | 64 | 9.2 | 4.0× | 19.6 |
| | Threshold Decoding | 70 | 8.1 | 3.7× | 41.6 |
| | Consistency Distill | 64 | 8.6 | 4.0× | 19.8 |
| | **SPEED (ours)** | **21** | **26.6** | **12.2×** | **41.2** |
| | **SPEED + JSD Decoding** | **21** | **25.1** | **12.2×** | **41.4** |

*Table 1.* Evaluation results on LLaDA-8B-Instruct. All runs use a maximum generation length of 256. We report the average number of decoding steps (model forward passes) and throughput in tokens/s, measured on an NVIDIA A4500 GPU.

and Consistency Distill (69.2%) in generation quality. On MATH, SPEED shortens the decoding steps from 256 to 36, increasing throughput from 1.8 to 17.9 tokens/s (7.1× speedup), while maintaining 32.0% accuracy. With JSD-based decoding, the model further improves to 32.8% accuracy at the cost of 3 additional steps on average, demonstrating its ability to balance efficiency and accuracy on more challenging reasoning tasks. Although the training data is sampled only from mathematical datasets, we observe similar trends on HumanEval and MBPP, suggesting strong generalization beyond the training distribution. On HumanEval, SPEED reaches 40.9% accuracy in 29 steps, and JSD decoding raises accuracy to 42.1% with only 2 additional steps, recovering most of the gap to greedy decoding (42.6%). On MBPP, SPEED achieves the highest speedup among all settings, reducing decoding steps to 21 and reaching a throughput of 26.6 tokens/s (12.2× speedup), with 41.2% accuracy. This result is close to the greedy baseline (42.4%) while outperforming all other accelerated baselines by a clear margin.

**Evaluation results on Dream.** Table 2 shows that our method delivers consistent improvements in decoding efficiency while preserving accuracy across all benchmarks

on Dream-7B-Instruct. On GSM8K, SPEED reduces the number of decoding steps from 256 to 38 and achieves a 6.7× speedup (26.6 tokens/s), with only a minor drop in accuracy from 79.3% to 78.4%. This outperforms all other accelerated baselines in both speed and accuracy. On MATH, a more challenging benchmark, SPEED reduces the steps to 71 and reaches 3.6× speedup while maintaining 39.7% accuracy, close to the greedy baseline (43.1%) and substantially outperforming Few-step and Consistency Distill, both of which suffer from sharp performance drops. With JSD decoding, accuracy further improves to 40.6% at a slight cost in speed, demonstrating the benefit of adaptive decoding under hard contexts. On HumanEval, SPEED achieves a 3.6× speedup and preserves the original model's accuracy of 47.6%. JSD decoding slightly improves accuracy to 47.7%, reinforcing its robustness on code generation. On MBPP, SPEED achieves 55.6% accuracy with a 4.7× speedup, closely matching the baseline (58.6%) while significantly outperforming Few-step and Consistency Distill in quality. Overall, the results confirm that SPEED generalizes well to Dream-7B-Instruct, and that both the distillation objective and the JSD-based decoding contribute to substantial efficiency gains with minimal quality degradation.

| Benchmark | Method | #Steps↓ | Tokens/s↑ | Speedup↑ | Accuracy↑ |
|---|---|---|---|---|---|
| GSM8K (5-shot) | Dream-7B-Instruct | 256 | 3.9 | 1.0× | 79.3 |
| | Few-step Decoding | 64 | 15.8 | 4.0× | 62.5 |
| | Threshold Decoding | 82 | 12.1 | 3.1× | 79.8 |
| | Consistency Distill | 64 | 15.5 | 4.0× | 64.7 |
| | **SPEED (ours)** | **38** | **26.6** | 6.7× | **78.4** |
| | **SPEED + JSD Decoding** | **41** | **24.2** | 6.2× | **78.4** |
| MATH (4-shot) | Dream-7B-Instruct | 256 | 2.8 | 1.0× | 43.1 |
| | Few-step Decoding | 64 | 12.4 | 4.0× | 18.7 |
| | Threshold Decoding | 127 | 5.1 | 2.0× | 43.2 |
| | Consistency Distill | 64 | 11.9 | 4.0× | 19.5 |
| | **SPEED (ours)** | **71** | **11.0** | 3.6× | **39.7** |
| | **SPEED + JSD Decoding** | **78** | **9.6** | 3.2× | **40.6** |
| HumanEval (0-shot) | Dream-7B-Instruct | 256 | 6.4 | 1.0× | 47.6 |
| | Few-step Decoding | 64 | 26.5 | 4.0× | 11.0 |
| | Threshold Decoding | 127 | 13.1 | 2.0× | 49.4 |
| | Consistency Distill | 64 | 25.9 | 4.0× | 14.2 |
| | **SPEED (ours)** | **72** | **24.2** | 3.6× | **47.6** |
| | **SPEED + JSD Decoding** | **76** | **22.7** | 3.4× | **47.7** |
| MBPP (3-shot) | Dream-7B-Instruct | 256 | 3.1 | 1.0× | 58.6 |
| | Few-step Decoding | 64 | 12.2 | 4.0× | 22.6 |
| | Threshold Decoding | 74 | 10.8 | 3.5× | 58.8 |
| | Consistency Distill | 64 | 11.9 | 4.0× | 23.1 |
| | **SPEED (ours)** | **55** | **14.8** | 4.7× | **55.6** |
| | **SPEED + JSD Decoding** | **60** | **13.9** | 4.3× | **55.6** |

*Table 2.* Evaluation results on Dream-7B-Instruct. All runs use a maximum generation length of 256. We report the average number of decoding steps (model forward passes) and throughput in tokens/s, measured on an NVIDIA A4500 GPU.

## 5.3. Ablation Study

We conduct ablation experiments on the LLaDA-8B-Instruct model to assess the impact of key design choices in our training objective. Results are summarized in Table 3.

**Configuration in the distillation loss.** To understand the roles of temperature scaling, we conduct training by removing either the student temperature $T_s$ or the teacher temperature $T_t$. Without $T_s$, the student distribution becomes overly sharp and less flexible, leading to slight degradation in both speedup and accuracy (9.1×, 77.6). On the other hand, removing $T_t$ while keeping $T_s$ leads to a softer teacher signal. This slightly improves accuracy (78.1) but increases the number of decoding steps to 37, reducing the speedup to 6.9×. These results highlight the necessity of using both sharpened teacher and softened student distributions to achieve an optimal balance between confidence calibration and training stability. We further examine a brute-force variant where the KLD loss is applied over all masked tokens in the active block $B_i$, rather than restricting it to the correctly predicted subset $\mathcal{M}_c$. Although this setting achieves the lowest number of decoding steps (21) and highest speedup (12.1×), it suffers from significant accuracy degradation (69.2). This indicates that forcing the model

to mimic uncertain or incorrect teacher predictions leads to harmful overconfidence.

| Settings | GSM8K (5-shot) | | |
|---|---|---|---|
| | #Steps↓ | Speedup↑ | Accuracy↑ |
| w/o $T_s$ | 28 | 9.1× | 77.6 |
| w/o $T_t$ | 37 | 6.9× | **78.1** |
| Entropy loss | 31 | 8.3× | 76.1 |
| Brute Distill | **21** | 12.1× | 69.2 |
| **SPEED** | 26 | 9.8× | 77.9 |

*Table 3.* Ablation study on different settings with the LLaDA-8B-Instruct model.

**Naive entropy loss.** An alternative way to sharpen model confidence is to directly minimize entropy during training. While this approach also improves decoding efficiency (31 steps, 8.3× speedup), it yields lower accuracy (76.1) compared to our KLD-based objective. This is because entropy minimization lacks an explicit alignment target and may over-penalize calibrated uncertainty. In contrast, the KLD loss aligns the student with confident and correct teacher predictions, making it more effective for learning selective confidence sharpening. These results confirm that our pro-

posed distillation design more reliably reduces decoding steps without compromising output quality.

**Sensitivity to training noise level.** The distillation in Eq. 9 draws its KL signal from the teacher-correct subset $\mathcal{M}_c$, whose size is governed by the masking ratio of the active block. To probe how this ratio shapes the distilled model, we retrain SPEED-LLaDA with the noise level $t$ fixed in $\{0.25, 0.5, 1.0\}$ and keep all other settings identical, then evaluate on GSM8K (5-shot). At $t = 0.25$ the teacher is almost always correct, yet $|\mathcal{M}_c|$ is too sparse to drive strong sharpening and the student converges to a less compact decoder. At $t = 1.0$ the block is fully masked and $|\mathcal{M}_c|$ peaks, but the teacher correct ratio collapses to $0.61$, so many KL targets are unreliable and the student commits to incorrect tokens. The intermediate $t = 0.5$ recovers both a high teacher correct ratio and a substantial $|\mathcal{M}_c|$, yielding the best accuracy at a competitive step count and justifying our choice in the main experiments.

| $t$ | Avg $|\mathcal{M}_c|$ | Correct Ratio | GSM8K↑ | #Steps↓ |
|-----|------|------|------|------|
| 0.25 | 7.96 | 0.99 | 73.5 | 31 |
| **0.5** | **15.85** | **0.99** | **77.9** | **26** |
| 1.0 | 19.57 | 0.61 | 72.4 | 23 |

*Table 4.* Sensitivity of SPEED-LLaDA to a fixed training noise level $t$ on GSM8K (5-shot). The intermediate level balances teacher correctness and $|\mathcal{M}_c|$, giving the strongest accuracy–efficiency trade-off.

| Method | CNN/DM | TruthfulQA | QASPER |
|--------|--------|-----------|--------|
| LLaDA-8B | 19.7 (1.0×) | 41.8 (1.0×) | 11.2 (1.0×) |
| **SPEED-LLaDA** | **20.8 (3.4×)** | **40.6 (7.1×)** | **10.5 (4.8×)** |

*Table 5.* Cross-domain evaluation of SPEED-LLaDA, reporting ROUGE for CNN/DailyMail, accuracy for TruthfulQA, and F1 for QASPER. Parentheses give the wall-clock speedup over greedy decoding.

**Generalization beyond reasoning and code.** Our training trajectories come entirely from mathematical prompts, so the gains on HumanEval and MBPP already hint at cross-domain transfer. To stress-test further, we evaluate SPEED-LLaDA on three task types outside the training distribution: CNN/DailyMail (Hermann et al., 2015) for abstractive news summarisation, TruthfulQA (Lin et al., 2022) for factual robustness against common misconceptions, and QASPER (Dasigi et al., 2021) for paper-grounded question answering. We hold the inference configuration fixed to the main-paper setting (256 tokens, block size 32, entropy threshold $\tau = 0.5$). As shown in Table 5, SPEED preserves task quality on all three benchmarks while delivering between $3.4\times$ and $7.1\times$ acceleration. The modest speedup on CNN/DailyMail reflects the lower inherent parallelisability of long narrative generation. Combined with the reasoning and code benchmarks in Tables 1 and 2, these numbers indicate that sharpened-teacher distillation transfers across qualitatively different generation regimes rather

than overfitting to mathematical reasoning.

## 6. Conclusion

We present SPEED, a training and inference framework for accelerating semi-autoregressive decoding in masked diffusion language models. By distilling from a sharpened teacher using a selective KL objective, the model learns to assign more decisive probabilities to tokens that are already predicted correctly. At inference time, we further introduce a JSD-based grouping strategy that dynamically separates context-sensitive tokens from those that can be decoded in parallel, enabling adaptive hybrid decoding. Experiments on two diffusion language models across mathematical reasoning and code generation tasks demonstrate that SPEED significantly reduces the number of decoding steps while preserving or even improving generation quality. Our approach is simple to implement and compatible with existing decoding frameworks, offering a practical solution for efficient text generation.

## Acknowledgement

This project is supported by the National Research Foundation, Singapore, and Cyber Security Agency of Singapore under its National Cybersecurity R&D Programme and CyberSG R&D Cyber Research Programme Office (Award: CRPO-GC1-NTU-002).

## Impact Statement

This paper presents a method for improving the efficiency of masked diffusion language models by reducing the number of decoding steps required during text generation. The primary goal of this work is to advance research in efficient large-scale language modeling and to make diffusion-based language models more practical for real-world deployment. By lowering computational cost and latency, our approach may broaden access to high-quality generative models in resource-constrained environments. At the same time, as with all advances in large language models, improved efficiency may facilitate wider deployment of text generation systems, which can have both positive and negative societal impacts. These include beneficial applications in education, programming assistance, and accessibility, as well as risks related to misinformation, misuse, or over-reliance on automated systems. Our method does not introduce new application domains or alter the fundamental capabilities of language models, but rather improves their inference efficiency. We encourage responsible deployment practices and adherence to existing safety, evaluation, and governance frameworks when applying such models in practice.

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

# A. Appendix

## A.1. Theoretical Justification for the JSD-based Constraint

**Theorem 1.** *Let the decoding process for a block $B_i$ be a partition into an ordered sequence of groups $\mathcal{P}_i = (G_1, \ldots, G_M)$. The error incurred at step $k$ due to the parallel decoding assumption is given by the KL divergence between the true sequential joint and the factorized approximation:*

$$E_{G_k} = \text{KL}\left( p_\theta(\boldsymbol{x}_{G_k}|\mathcal{C}_k) \,\middle\|\, \prod_{j \in G_k} p_\theta(x_j|\mathcal{C}_k) \right), \tag{12}$$

*where $\mathcal{C}_k = (\boldsymbol{x}_{B_{<i}}, \boldsymbol{x}_{G_{<k}}, \ldots)$ is the full context available before decoding group $G_k$. The total error for the block is $E_{total} = \sum_{k=1}^{M} E_{G_k}$.*

*The JSD score for a token $j \in B_i$, defined as $\mathcal{J}_j = \text{JSD}\left(p_\theta(\cdot|\mathcal{C}_1) \,\|\, p_\theta(\cdot|\mathcal{C}_0)\right)$, where $\mathcal{C}_1$ is the context with the true past block $\boldsymbol{x}_{B_{<i}}$ and $\mathcal{C}_0$ is the context with it masked, quantifies the token's sensitivity to past-block context. A decoding strategy that applies a more conservative grouping (i.e., smaller group sizes) to tokens with higher $\mathcal{J}_j$ scores serves as a principled approach to minimizing the total expected generation error $E_{total}$.*

The proof proceeds in three parts. First, we decompose the group error term $E_{G_k}$ to reveal its dependence on intra-group conditional information. Second, we relate the JSD metric to information-theoretic quantities that measure contextual sensitivity. Finally, we argue that high contextual sensitivity, as measured by JSD, implies a higher expected contribution to the error term, justifying the proposed constrained grouping strategy.

**1. Decomposing the Parallelization Error.** The error term $E_{G_k}$ quantifies the discrepancy introduced by ignoring the dependencies among tokens within the group $G_k$. Using the chain rule for probability on the true joint, $p_\theta(\boldsymbol{x}_{G_k}|\mathcal{C}_k) = \prod_{j \in G_k} p_\theta(x_j|\boldsymbol{x}_{G_k,<j}, \mathcal{C}_k)$, where $< j$ denotes an arbitrary but fixed ordering within the group. The KL divergence can be expanded as follows:

$$E_{G_k} = \mathbb{E}_{\boldsymbol{x}_{G_k} \sim p_\theta(\cdot|\mathcal{C}_k)} \left[ \log p_\theta(\boldsymbol{x}_{G_k}|\mathcal{C}_k) - \log \prod_{j \in G_k} p_\theta(x_j|\mathcal{C}_k) \right] \tag{13}$$

$$= \mathbb{E}_{\boldsymbol{x}_{G_k} \sim p_\theta(\cdot|\mathcal{C}_k)} \left[ \sum_{j \in G_k} \log p_\theta(x_j|\boldsymbol{x}_{G_k,<j}, \mathcal{C}_k) - \sum_{j \in G_k} \log p_\theta(x_j|\mathcal{C}_k) \right] \tag{14}$$

$$= \sum_{j \in G_k} \mathbb{E}_{\boldsymbol{x}_{G_k,\leq j} \sim p_\theta(\cdot|\mathcal{C}_k)} \left[ \log \frac{p_\theta(x_j|\boldsymbol{x}_{G_k,<j}, \mathcal{C}_k)}{p_\theta(x_j|\mathcal{C}_k)} \right] \tag{15}$$

$$= \sum_{j \in G_k} \mathbb{E}_{\boldsymbol{x}_{G_k,<j} \sim p_\theta(\cdot|\mathcal{C}_k)} \left[ \text{KL}\left( p_\theta(\cdot|\boldsymbol{x}_{G_k,<j}, \mathcal{C}_k) \,\|\, p_\theta(\cdot|\mathcal{C}_k) \right) \right]. \tag{16}$$

This decomposition shows that the total error for a group is the sum of expected KL divergences. Each term represents the information gained about a token $x_j$ from knowing the other tokens decoded just before it within the same parallel step. The parallel decoding error is large if tokens within a group strongly inform one another.

**2. The JSD as a Measure of Contextual Sensitivity.** The Jensen-Shannon Divergence between the predictive distributions for token $j$ under context $\mathcal{C}_1$ (past revealed) and $\mathcal{C}_0$ (past masked) is defined as:

$$\mathcal{J}_j = \frac{1}{2}\text{KL}(p_1\|p_M) + \frac{1}{2}\text{KL}(p_0\|p_M), \tag{17}$$

where $p_1 = p_\theta(\cdot|\mathcal{C}_1)$, $p_0 = p_\theta(\cdot|\mathcal{C}_0)$, and $p_M = \frac{1}{2}(p_1 + p_0)$ is the mixture distribution. The JSD is the mutual information between the random variable for token identity $X_j$ and a binary random variable $C$ representing the context choice ($C = 0$ for $\mathcal{C}_0$, $C = 1$ for $\mathcal{C}_1$). A high $\mathcal{J}_j$ signifies that revealing the past context provides substantial information about the identity of token $j$, implying that the token's predictive distribution is highly sensitive to its surrounding context. Such tokens are often linguistically pivotal, resolving significant ambiguity in the sequence.

**3. Linking Contextual Sensitivity to Parallelization Error.**   The core of our argument rests on the well-founded linguistic assumption that a token's sensitivity to its context is a general property. A token whose identity is highly uncertain without the preceding block's context (high $\mathcal{J}_j$) is also likely to be one whose identity is highly uncertain without the context provided by its peer tokens within a decoding group. This is because both contexts serve to resolve ambiguity.

Let us consider a token $j$ with a high JSD score, $\mathcal{J}_j$. This indicates that its predictive distribution $p_\theta(x_j|\cdot)$ is highly variable with respect to changes in the conditioning context. When such a token is placed in a large parallel group $G_k$, it is plausible that the information provided by its peer tokens $\boldsymbol{x}_{G_k,<j}$ would also cause a significant shift in its distribution. This leads to a large value for the corresponding KL term in the error decomposition (Eq. 16).

$$\mathbb{E}_{\boldsymbol{x}_{G_k,<j}} \left[ \text{KL} \left( p_\theta(\cdot \mid \boldsymbol{x}_{G_k,<j}, \mathcal{C}_k) \parallel p_\theta(\cdot \mid \mathcal{C}_k) \right) \right] \text{ is expected to be large if } \mathcal{J}_j \text{ is large.} \tag{18}$$

Consequently, including tokens with high JSD scores in large parallel groups is likely to contribute disproportionately to the total generation error $E_{\text{total}}$.

Our proposed strategy directly mitigates this risk. By partitioning tokens into a "slow set" $\mathcal{S}_{\text{slow}}$ (high JSD) and a "fast set" $\mathcal{S}_{\text{fast}}$ (low JSD), we isolate the high-risk tokens. Applying a conservative decoding strategy (e.g., high confidence threshold $\tau_{\text{high}}$, leading to small or singleton groups) to $\mathcal{S}_{\text{slow}}$ ensures that these sensitive tokens are decoded with more complete context, thereby minimizing their contribution to the parallelization error. Conversely, for tokens in $\mathcal{S}_{\text{fast}}$, their low JSD suggests robustness to contextual variations, making the factorized approximation in Eq. 6 more accurate and justifying an aggressive parallelization strategy. This hybrid approach thus provides a principled method for managing the speed-quality trade-off by allocating computational caution where it is most needed, thereby minimizing the total expected error.

## A.2. Empirical Diagnostics for Safe Parallel Decoding

This subsection details the two complementary diagnostics plotted in Figure 1, namely the intra-group dependency error probe in panel (a) and the safely-jointly-decodable tokens probe in panel (b). We document the construction of each so that the reported curves can be reproduced.

**Proxy for intra-group dependency error.**   For every question–answer pair in the GSM8K test split, we first obtain a greedy completion under semi-autoregressive decoding from both LLaDA-8B-Instruct and SPEED-LLaDA, which serves as a per-model pseudo-label. We then apply semi-autoregressive masking to each generated trajectory, keeping the preceding blocks $\boldsymbol{x}_{B_{<i}}$ intact and masking the current and subsequent blocks $\boldsymbol{x}_{B_{\geq i}}$. The first active block $B_i$ is re-denoised in two ways, namely a single-token-per-step greedy pass that serves as the reference and a fixed-group-size pass that unmasks $|G|$ tokens simultaneously at every step. We compute the token-wise agreement ratio on $B_i$ between the two traces and average it across all evaluation samples. A lower agreement reflects a larger violation of the conditional independence assumption when $|G|$ tokens are committed together. Figure 1(a) sweeps $|G| \in \{1, 2, 4, 8, 16, 32\}$ and reports the mean self-consistency, with the corresponding numerical values listed in Table 6. SPEED retains higher consistency at every $|G|$, which directly quantifies the reduced intra-group dependency error that the sharpened student delivers.

| $|G|$ | | 1 | 2 | 4 | 8 | 16 | 32 |
|---|---|---|---|---|---|---|---|
| LLaDA-8B-Instruct | | 1.000 | 0.882 | 0.803 | 0.748 | 0.708 | 0.678 |
| **SPEED-LLaDA** | | **1.000** | **0.905** | **0.825** | **0.781** | **0.728** | **0.691** |

*Table 6.* Self-consistency at fixed parallel group size $|G|$ on GSM8K, used to plot panel (a) of Figure 1.

**Proxy for safely jointly decodable tokens.**   At the first decoding step of an active block, a confidence threshold $\tau$ defines the set of tokens that the model deems safe to commit in parallel. We measure the size of this set directly by counting, for every block in the GSM8K test split, how many of the masked tokens already satisfy a confidence threshold $\tau$ at step one, and we average the count across the test split. Figure 1(b) reports this average as $\tau$ sweeps from 0.4 to 0.9, with numerical values listed in Table 7. Across the full range, SPEED produces 2–4 additional safe tokens per block compared with the base model, which translates into wider parallel groups under threshold decoding and is the direct mechanism behind the reduced step counts reported in Tables 1 and 2.

**Discussion.**   The two diagnostics measure complementary aspects of the same mechanism. The first quantifies the proxy intra-group dependency error that bounds the quality of parallel generation under controlled fixed-group decoding, while

| $\tau$ | 0.4 | 0.5 | 0.6 | 0.7 | 0.8 | 0.9 |
|---|---|---|---|---|---|---|
| LLaDA-8B-Instruct | 18.4 | 17.1 | 15.7 | 14.4 | 12.9 | 11.0 |
| **SPEED-LLaDA** | **20.5** | **19.7** | **18.9** | **18.1** | **17.1** | **15.6** |

*Table 7.* Average number of safely jointly decodable tokens per block at the first decoding step on GSM8K, used to plot panel (b) of Figure 1.

the second quantifies the safe-set size that threshold decoding can actually exploit at inference. A lower dependency error is what makes larger groups safer, and a larger safe set is what allows the decoder to take those larger groups in practice. SPEED improves both quantities simultaneously, which provides direct empirical evidence that the sharpened-teacher distillation in Section 4.1 achieves the goal stated in Section 3 of enlarging the set of jointly decodable tokens without sacrificing correctness.

### A.3. Limitation

Despite its effectiveness, our approach has several limitations. First, the training procedure relies on high-quality teacher-generated trajectories and filtered correct responses, which introduces additional computational cost and may limit applicability when reliable supervision is unavailable. Second, the JSD-based strategy introduces extra computation during inference due to additional forward evaluations for dependency estimation, partially offsetting the raw speed gains in certain settings. Moreover, the adaptive grouping strategy is tuned using fixed thresholds and block sizes, which may not be optimal across model scales or domains. Finally, while we demonstrate strong performance on reasoning and code benchmarks, broader evaluation on open-ended generation and long-form tasks is needed to fully assess generalization and robustness.

