# OpenReview forum: "SPEED: Sharpened-Teacher Distillation for Parallel Decoding of Diffusion Language Models"
_ICML.cc/2026/Conference — ICML 2026 regular_

### Official Review · Reviewer_SJEH · 2026-03-09

**Soundness:** 3
**Presentation:** 3
**Significance:** 3
**Originality:** 3
**Overall Recommendation:** 4
**Confidence:** 2

**Summary:**

This paper introduces SPEED, a framework that accelerates Masked Diffusion Models by using Sharpened-Teacher Distillation to increase token prediction confidence and a Slow-Fast Decoding strategy to adaptively manage token dependencies via JS Divergence. It achieves up to 12.2× speedup on LLaDA-8B while maintaining high generation quality, effectively balancing efficiency and robustness in parallel decoding.

**Compliance With Llm Reviewing Policy:**

Affirmed.

**Final Justification:**

My concerns are addressed. Considering the novelty and contribution of the paper, I tend to keep my scores unchanged.

**Key Questions For Authors:**

N/A

**Limitations:**

yes

**Strengths And Weaknesses:**

# Strengths
1. SPEED achieves a remarkable speedup (up to 12.2× on LLaDA-8B) while maintaining generation quality close to greedy decoding. It effectively addresses the "speed-quality trade-off" that has long plagued parallel decoding in diffusion language models.
2. The introduction of Slow-Fast Decoding based on Jensen-Shannon Divergence (JSD) provides a theoretically sound and dynamic way to manage token dependencies. Instead of static block sizes, it intelligently identifies which tokens are safe to decode in parallel and which require more context.

# Weaknesses
1. The method requires high-quality teacher trajectories for distillation, which increases training costs. Additionally, the JSD calculation requires extra forward passes during inference to estimate dependencies。
2. While the paper demonstrates strong results on reasoning and coding tasks (like HumanEval and GSM8K), its robustness in open-ended generation and long-form tasks has not been fully explored, leaving questions about its performance in more creative or less-structured linguistic contexts.

---

> ### Author Rebuttal · Authors · 2026-03-30
>
> Thanks for the insightful and positive feedback. We have provided detailed responses below and hope they address your concerns. We really appreciate your time and consideration.
>
> > **W1-1: Extra training cost.**
>
> Thanks for raising this point. We agree that distillation introduces additional compute due to generating teacher trajectories. In our current implementation, we generate these trajectories offline once and store them on disk, so the distillation stage reuses a fixed dataset rather than repeatedly sampling trajectories online. We will explicitly report this one-time data generation cost and discuss it as a limitation.
>
> > **W1-2: Extra forward passes with JSD.**
>
> Thanks for pointing this out. Our JSD decoding implementation does not require extra model forward passes to estimate dependencies. Under semi-autoregressive decoding, we already obtain logits for (i) the first step when decoding block $B_i$ (where $B_i$ is fully masked) and (ii) the first step when decoding block $B_{i+1}$ (where $B_i$ is fully decoded). We reuse these existing logits to compute token-wise JSD scores for $B_{i+1}$, so the dependency estimate is effectively free in terms of additional model evaluations. The main overhead of enabling JSD is indirect: the fast/slow partition can introduce a small number of extra decoding groups (hence a few more decoding steps) within a block, trading a bit of speed for improved robustness/accuracy. We will clarify this implementation detail and the overhead/benefit trade-off in the revision.
>
> >  **W2: Evaluations on other tasks.**
>
> Thanks for the reviewer’s interest in robustness beyond code and math reasoning. To evaluate performance in less-structured linguistic contexts, we additionally test on three diverse benchmarks: CNN/DailyMail [1] (abstractive news summarization), TruthfulQA [2] (truthfulness-focused QA designed to elicit common misconceptions), and QASPER [3] (paper-based QA that requires answering information-seeking questions conditioned on full scientific documents).
>
> We evaluate SPEED-LLaDA with max generation length 256, block size 32, and the same entropy-threshold decoding as in the main paper (threshold $0.5$). Results are:
>
> | Method         |   CNN/DailyMail    |     TruthfulQA     |       QASPER       |
> | :------------- | :----------------: | :----------------: | :----------------: |
> | Original LLaDA | 19.7 (1.0$\times$) | 41.8 (1.0$\times$) | 11.2 (1.0$\times$) |
> | SPEED-LLaDA    | 20.8 (3.4$\times$) | 40.6 (7.1$\times$) | 10.5 (4.8$\times$) |
>
> These results suggest that SPEED maintains comparable task performance while providing substantial acceleration on summarization and open-ended QA settings, not only on reasoning/coding benchmarks. We will include these additional evaluations in the revision to better characterize robustness across task domains.
>
> [1] "Teaching machines to read and comprehend." Advances in neural information processing systems, 2015
>
> [2] "Truthfulqa: Measuring how models mimic human falsehoods." In Proceedings of the 60th annual meeting of the association for computational linguistics, 2022.
>
> [3] "A dataset of information-seeking questions and answers anchored in research papers." In Proceedings of the 2021 Conference of the North American Chapter of the Association for Computational Linguistics, 2021.

---

> > ### Author Rebuttal · Reviewer_SJEH · 2026-04-07
> >
> > The authors' rebuttal has resolved my concerns.

---

### Official Review · Reviewer_NXpH · 2026-03-12

**Soundness:** 3
**Presentation:** 2
**Significance:** 2
**Originality:** 2
**Overall Recommendation:** 4
**Confidence:** 3

**Summary:**

This paper addresses the speed-quality tradeoff in parallel decoding for masked diffusion language models (MDMs). The authors frame MDM decoding as an iterative token grouping problem and argue that quality degradation at aggressive parallelism stems from grouping interdependent tokens together, violating the conditional independence assumption. To mitigate this, they propose SPEED, a framework with two complementary components. On the training side, a sharpened teacher distillation objective applies a temperature-scaled KL divergence loss selectively on positions where the teacher already predicts correctly, encouraging the student to concentrate probability mass on correct tokens and push more predictions above the decoding threshold. On the inference side, a Slow-Fast Decoding strategy uses token-wise Jensen-Shannon Divergence to partition tokens into a context-resolved "fast" set that can be decoded aggressively in parallel and a context-sensitive "slow" set that is decoded more cautiously. Experiments on LLaDA-8B-Instruct and Dream-7B-Instruct across GSM8K, MATH, HumanEval, and MBPP show speedups of up to 12.2x with minimal accuracy loss compared to greedy decoding, substantially outperforming baselines.

**Compliance With Llm Reviewing Policy:**

Affirmed.

**Final Justification:**

The authors have addressed my concerns. I am now more confident in the merits of this work and have decided to raise my score accordingly.

**Key Questions For Authors:**

The selective KL loss is applied only to positions where the teacher predicts correctly (Mc). How sensitive is training to the size of this set? At high noise levels, Mc may be very small, making the KL signal sparse; at low noise levels, Mc may cover nearly all tokens, making it resemble brute-force distillation. A breakdown of Mc size across different noise levels t and its impact on final performance would help clarify the robustness of this design choice.

**Limitations:**

Yes

**Strengths And Weaknesses:**

Strengths:

1. The framing of MDM decoding as an iterative token grouping problem is insightful. It provides a lens through which to understand why lowering the confidence threshold degrades quality, namely that interdependent tokens get forced into the same parallel group. The JSD-based partitioning into slow and fast sets follows naturally from this framing and is well motivated both empirically and theoretically. The overall narrative from problem diagnosis to solution is coherent and easy to follow.

2. The experimental results are strong across both models and four benchmarks. SPEED achieves large speedups (up to 12.2x on MBPP) while keeping accuracy close to greedy decoding, and it consistently outperforms few-step decoding and consistency distillation, both of which suffer catastrophic quality drops. The ablation study in Table 3 is informative, clearly isolating the contribution of each design choice (teacher temperature, student temperature, selective vs. brute-force distillation).



Weaknesses:

The JSD-based Slow-Fast Decoding requires one additional forward pass per block (with the preceding block masked out) to compute the sensitivity partition. Across all benchmarks, this extra cost consistently reduces throughput compared to SPEED alone (e.g., 16.1 to 14.8 tokens/s on GSM8K for LLaDA), while the accuracy gains are marginal (typically 0.1 to 0.9 points). The step count sometimes even increases slightly. Given this unfavorable cost-benefit tradeoff, the practical value of the JSD component on top of the distillation is not convincingly established, and one could argue that the main contribution is really just the sharpened teacher distillation.

---

> ### Author Rebuttal · Authors · 2026-03-30
>
> Thank you for your insightful comments! Below, we provide a point-by-point response to address your concerns. We welcome further discussion to improve the clarity and effectiveness of our work.
>
> > **W1: Cost–benefit tradeoff for the JSD-based decoding.**
>
> We appreciate your careful analysis. We acknowledge that JSD-based decoding may **increase decoding steps**, but this increase does not come directly from computing the JSD. We also find that the cost–benefit tradeoff can be improved with a better thresholding strategy for partitioning the slow and fast sets.
>
> (1) No additional forward passes for JSD computing; extra steps come from deliberate grouping
>
> In our implementation, computing JSD does not require an additional forward pass: we reuse logits that are already produced at the first decoding step of adjacent blocks, and compute the JSD values directly from these cached logits. Instead, the step count increases because JSD decoding with mean threshold enforces a partition: high-sensitivity tokens are decoded in parallel (fast set), while low-sensitivity tokens are deferred to reduce error propagation, which naturally raises the number of decoding groups (and thus steps) modestly.
>
> (2) Performance is bottlenecked by the thresholding strategy
>
> The original mean-threshold based JSD decoding  can yield an unfavorable cost–benefit tradeoff because it **forces a split in every block** regardless of whether tokens are meaningfully separable by sensitivity. Since Jensen–Shannon divergence (JSD) is bounded in $[0,\ln 2]$, a fixed absolute cutoff is more intuitive and can avoid over-partitioning. We therefore evaluate an updated variant using a fixed cutoff of $0.3\ln 2$ to partition tokens into slow/fast sets on SPEED-LLaDA. Results below report accuracy (avg steps):
>
> | Method | GSM8K | HumanEval | MATH | MBPP |
> |:--|:--:|:--:|:--:|:--:|
> | Original LLaDA | 79.3 (256) | 42.6 (256) | 33.5 (256) | 42.4 (256) |
> | SPEED-LLaDA | 77.9 (26) | 40.9 (29) | 32.0 (36) | 41.2 (21) |
> | SPEED-LLaDA + JSD (mean-split) | 78.3 (28) | 41.0 (29) | 32.8 (38) | 41.4 (21) |
> | SPEED-LLaDA + JSD* (fixed cutoff) | 78.9 (29) | 42.1 (31) | 32.8 (39) | 41.4 (21) |
>
> With the fixed-cutoff rule, GSM8K improves from 77.9 to 78.9 with +3 steps on average, and HumanEval improves from 40.9 to 42.1 with +2 steps, showing a more favorable robustness–overhead tradeoff than our initial mean-based split. We will revise the paper to present these analyses and the updated JSD decoding as a meaningful optional component that prudently trades a small overhead for improved accuracy.
>
>
>
> > **Q1: Training sensitivity to noise level.**
>
> Thanks for the thoughtful question. We agree that the selective KL signal depends on the size of $M_c$, which is strongly correlated with the noise level $t$. To study robustness, we run ablation experiments on LLaDA-8B-Instruct with three fixed noise levels $t\in\{0.25, 0.5, 1.0\}$ (all other settings unchanged). During training, we log the average $|M_c|$ and the teacher correct ratio, which is  the number of correctly predicted masked tokens divided by the number of masked tokens in the active block. After training, we evaluate the resulting models on GSM8K (5-shot) with the same entropy-threshold decoding. Results are shown in following table,
>
> | Noise Level | Avg $\|M_c\|$ | Correct Ratio | GSM8K Score | Avg Steps |
> |:--:|:--:|:--:|:--:|:--:|
> | 0.25 | 7.96 | 0.9907 | 73.5 | 31 |
> | 0.5  | 15.85 | 0.9878 | 77.9 | 26 |
> | 1.0  | 19.57 | 0.6115 | 72.4 | 23 |
>
> These results highlight the intended trade-off. At low noise ($t=0.25$), $M_c$ is small: the KL signal becomes sparse (even though highly accurate), which limits confidence sharpening and leads to slower decoding and lower GSM8K accuracy. At high noise ($t=1.0$), $|M_c|$ becomes larger but the teacher correct ratio drops sharply, so the KL targets are less reliable and performance degrades despite fewer steps. The intermediate noise level ($t=0.5$) yields both a high teacher correct ratio and a moderate-but-sufficient $|M_c|$, producing the best accuracy–efficiency balance, which justify our design choice of using fixed $t=0.5$ in main experiments.
> We will include this ablation study and discussion  in the revision.

---

> > ### Author Rebuttal · Reviewer_NXpH · 2026-04-03
> >
> > Thanks to the authors for their thorough rebuttal. My concerns have been addressed, and after reading the other reviewers' comments, I am now more confident in the contribution of this work. I will raise my score accordingly.

---

> > > ### Author Response · Authors · 2026-04-04
> > >
> > > We sincerely thank the reviewer for the thoughtful reconsideration. We are glad our rebuttal fully addressed your concerns and will further revise the manuscript in the final version.

---

### Official Review · Reviewer_Np5m · 2026-03-13

**Soundness:** 3
**Presentation:** 3
**Significance:** 3
**Originality:** 3
**Overall Recommendation:** 4
**Confidence:** 4

**Summary:**

This paper studies how to improve the speed-quality tradeoff of semi-autoregressive decoding in masked diffusion language models. The key idea is to view decoding as a token-grouping problem: quality degrades when aggressive parallel decoding groups together tokens with strong unresolved dependencies. To address this, the paper proposes SPEED, which combines a sharpened teacher distillation objective and a JSD-based Slow-Fast decoding. Experiments on LLaDA-8B-Instruct and Dream-7B-Instruct show substantial speedups with relatively small quality degradation.

**Compliance With Llm Reviewing Policy:**

Affirmed.

**Key Questions For Authors:**

1.	Can the authors provide a more direct diagnostic for the paper’s main mechanism? In particular, can they show that SPEED reduces some explicit proxy of intra-group dependency error, or increases the number of safely jointly decodable tokens?
2.	Why should high-JSD tokens be decoded more aggressively rather than more conservatively? A clearer conceptual or empirical justification here would materially improve my confidence in the JSD component.
3.	How much of the gain comes from selective distillation itself, versus temperature sharpening, versus JSD-based grouping? The current evidence suggests the first component is doing most of the work.
4.	How sensitive is the method to the use of teacher-generated, correctness-filtered training data? Would the approach still work under noisier or less curated supervision?

**Limitations:**

The paper briefly acknowledges broader societal implications of more efficient language generation, which is good. However, the technical limitations are under-discussed. In particular, the paper should be more explicit that the mechanism is not directly validated, that the JSD contribution is comparatively modest, and that the evidence is concentrated on teacher-filtered reasoning/code settings.

**Strengths And Weaknesses:**

Strengths:

1.	The paper identifies a real bottleneck and proposes a simple training-plus-inference recipe that appears effective.
2.	The overall framing of decoding as token grouping is helpful to provide a coherent motivation.
3.	Empirically, the results are strong enough to establish practical value with high efficiency.

Weaknesses:

1.	The paper is more convincing as an empirical speedup method than as a mechanistic explanation. The claim that aggressive parallel decoding fails mainly because of suboptimal grouping and conditional-independence violations is plausible, but it is not directly validated. The paper shows that the method works, but does not clearly demonstrate that it reduces groupwise dependency error or increases the number of safely jointly decodable tokens.
2.	The rationale for the JSD-based decoding rule is not fully clear. The paper places high-JSD tokens in the fast set and treats low-JSD tokens more conservatively, but the intuition for this choice is not well explained. The distinction between dependence on previous-block context and intra-block dependence should be clarified. Moreover, the gains from JSD over SPEED alone are fairly small in the main results, so its role currently appears incremental rather than central.
3.	The scope of empirical evidence is somewhat narrow. The training data are teacher-generated and filtered to retain only final-answer-correct samples, and the evaluation is limited to reasoning and code tasks. This is a reasonable starting point, but it makes the broader claims harder to assess. The paper would be stronger with either more restrained claims or broader validation and stronger diagnostics.

---

> ### Author Rebuttal · Authors · 2026-03-30
>
> Thanks for the insightful and valuable feedback. We have provided detailed responses below and hope they address your concerns. We really appreciate your time and consideration.
>
> >**W1 & Q1: Intra-group dependency error / safely jointly decodable tokens.**
>
> Thanks for the reviewer's interest.  We report two proxies to explicitly reflect intra-group dependency error and the number of safely jointly decodable tokens.
>
> (1) Proxy for intra-group dependency error. We measure *self-consistency* by comparing token-wise decoding outcomes between (a) greedy intra-block decoding (decode top-1 token per step) and (b) fixed group-size decoding (decode top-$|G|$ tokens per step).
>
> (2) Proxy for safely jointly decodable tokens.  We define *safely jointed decodable tokens* in a fully-noisy active block $B_i$ as masked positions whose predicted token is (a) identical to the pseudo target within that block and (b) has confidence above a threshold $\tau$.
>
> Both experiments results are shown in the anonymous link  https://anonymous.4open.science/r/Rebuttal-6901 .
>
>
> > **W2 & Q2: Rationale for JSD-based decoding.**
>
> We agree the intuition behind the JSD-based decoding should be explained more clearly. Our intent is to separate two different sources of uncertainty inside a block: (i) dependence on the already-generated previous blocks, and (ii) dependence on still-masked tokens within the same block. JSD is computed by toggling whether the previous block is revealed, so it primarily measures how much a token’s distribution is determined once cross-block context is available. A high JSD means the token becomes strongly constrained after seeing the previous block. In practice these are the “context-resolved” tokens that the model tends to decode early even under greedy decoding, which is exactly what Fig. 1 shows. Decoding them aggressively is beneficial because generating them early provides additional concrete context that helps resolve the remaining. Conversely, low-JSD tokens are more likely dominated by intra-block dependencies, so decoding them conservatively reduces the risk of grouping mutually dependent tokens together.
>
> We also verify that the grouping direction is essential: reversing the rule (decoding high-JSD tokens last) degrades both accuracy and efficiency compared to the normal direction, results are also shown in the same link. Regarding the small gains over SPEED in the main tables, we view JSD as a robustness layer whose benefit is most apparent on harder settings and/or more permissive thresholds. We also report an improved cost–benefit configuration for JSD decoding in our response to reviewer NXpH.
>
>
> > **W3: More constrained claims or broader validation.**
>
> To broaden validation beyond reasoning and code, we additionally evaluate SPEED-LLaDA on three diverse benchmarks. The results (reported in our reply to reviewer SJEH) show substantial speedups with only marginal performance changes. At the same time, we agree our current study starts from teacher-generated, correctness-filtered trajectories, and we will refine the paper’s claims to be more precise.
>
> > **Q3: More detailed component-wise contribution.**
>
> Thanks for the reviewer's interest. Actually, selective distillation alone is not sufficient and can even hurt speed when used without sharpening. To validate, we remove temperature sharpening by setting $T_t=T_s=1.$, the distilled LLaDA achieves 79.1 accuracy on GSM8K but requires 80 average steps under the same confidence-based decoding, compared to 78.7 accuracy and 77 steps for the original LLaDA. This indicates that the main efficiency gain comes from combination of selective distillation and temperature sharpening , while JSD-based grouping mainly provides a smaller robustness/quality benefit in harder or more permissive decoding regimes.
>
>
> > **Q4: Sensitivity to noisy training data.**
>
> Thanks for raising the sensitivity to noise in training data. We ran a controlled noise injection study by replacing 40K samples in our cleaned LLaDA distillation set (around 120K total) with previously filtered Numina-Math trajectories that have incorrect final answers, then retraining with the same setup and evaluating GSM8K (5-shot) under the same entropy-threshold decoding. The model trained with noisier supervision achieves 75.2 accuracy with 29 average steps, compared to 77.9 accuracy with 26 average steps when trained on the fully cleaned 120K data. This shows the approach remains effective under moderate noise (speedup persists), but both accuracy and efficiency degrade when incorrect trajectories are introduced. We believe this happens because wrong final answers often reflect trajectories where the teacher’s intermediate token predictions are unreliable, so selectively sharpening on “teacher-correct” positions becomes less informative and can also shift the student toward less stable decoding behavior. We will include this ablation and the discussion in the revision.

---

> > ### Author Rebuttal · Reviewer_Np5m · 2026-04-04
> >
> > Thank you for the detailed rebuttal. The authors have addressed my concerns.

---

> > > ### Author Response · Authors · 2026-04-04
> > >
> > > Thank you very much for your recognition of our work. We sincerely appreciate your constructive comments, which have been highly valuable in improving our manuscript. We will carefully revise the manuscript based on our disscussion.

---

### Official Review · Reviewer_UQU5 · 2026-03-14

**Soundness:** 4
**Presentation:** 3
**Significance:** 3
**Originality:** 2
**Overall Recommendation:** 4
**Confidence:** 5

**Summary:**

This paper presents SPEED, a framework designed to safely balance efficient parallel decoding and generation quality in MaskLM-based dLLMs. SPEED solves the tradeoff through complementary training and inference designs. During training, sharpened-teacher distillation teaches the student model to concentrate probability mass on correctly predicted tokens. At inference, a Jensen-Shannon Divergence (JSD) strategy partitions tokens based on their sensitivity to generated context, decoding stable tokens in parallel, while carefully deferring ambiguous ones. This combined approach achieves up to 12.2x and 6.7x speedups on LLaDA-8B-Instruct and Dream-7B-instruct, maintaining Acc% close to greedy decoding.

**Compliance With Llm Reviewing Policy:**

Affirmed.

**Final Justification:**

I'd like to hold my opinion of minor supporting the acceptance. My previous misunderstanding is mostly around the cost to transfer the successful results across domain and task, having the worry that the author(s) could have been merely meta-overfitting the benchmarks using hyperparameters. This seemed like a major concern/misunderstanding across my fellow reviewers too. I think the rebuttal addressed my concern, but not yet pushing it to the next level of confidence in supporting it, as obviously the clarity in writing on this matter can still be improved.

**Key Questions For Authors:**

I have only one core question: do you think such acceleration is generalizable across domains where the parallelism of generation is naturally different (translation against storytelling, for example), or this is indeed a problem that the current method is also struggling. In the worst case, can SPEED be even slower than simply performing sequential (one masked token each time) out-of-order decoding?

**Limitations:**

Yes.

**Strengths And Weaknesses:**

- Soundness:
 - The authors identify a clear, motivated cause for performance degradation in parallel decoding, caused by the violation of the conditional independence assumption when interdependent tokens are generated together. The authors provide theoretical grounding, evidences for empirical effectiveness and validation on design choices to fully analyse and tackle this issue. This idea also shares many aspects of previous adpatively parallel decoding post-train recipes that layerizes the tokens for safely parallelized decoding, such as InsNet-Dinic (https://papers.neurips.cc/paper_files/paper/2022/file/2e32d3a10985fc94c7e11ee6ea165cca-Paper-Conference.pdf).
 - My major concerns would still be on the high training cost and data reliance. Just like those previous methods, such decoding oriented post-training is usually expensive, and can be highly task-specific and non-generalizable. This is due to the high variance of parallelizability amongst different generation tasks, and unless actively merged into the pretraining stage of future large-scale dLLMs, I can't think of a good way to suggest for mitigating this issue. Also, compared to previous method that tries to directly record the preferences of parallelism into parameter, if I understand it correctly this paper's method would still require extra forward passes to safeguard the parallelization of token generation, which is a little bit disappointing.

- Presentation: The presentation is clear and conducted with a solid theoretical structure. The notation and proofs reveal the detailed theoretical essence of the problem, and are fairly easy to follow.

- Significance: Yes, the parallelism-quality trade-off has always been one of the most important problems in dLLMs. This paper pushes the community's understanding of this problem one step forward.

- Originality: While the idea of using grouping/self-distilling to safely identify parallelization in token generation with dLLMs is not new, the perspective and theory this paper presents still has its value. It's fairly novel.

---

> ### Author Rebuttal · Authors · 2026-03-30
>
> Thank you for your insightful and positive comments! Below, we provide a point-by-point response to address your concerns. We welcome further discussion to improve the clarity and effectiveness of our work.
>
> > **W1-1: High training cost and data reliance.**
>
> We appreciate this concern. We agree that our decoding-oriented post-training introduces additional cost, but it also significantly improves the speed–quality tradeoff frontier for diffusion LLMs.
>
> - First, the practical data and compute overhead is modest. The ~90K (Dream) / 120K (LLaDA) training pairs are generated offline once and stored on disk, so distillation reuses a fixed dataset rather than repeatedly sampling trajectories online. Training takes about 6 hours on 4 H200 GPUs.
>
> - Second, the learned capability is not task-specific and generalizes across tasks. Although our training data is sampled only from math datasets, we observe consistent speedup on code benchmarks (HumanEval/MBPP) in Tables 1 and Table 2. This shows the method improves early-step confidence calibration and safer parallel grouping, rather than memorizing task-specific patterns.
>
> We acknowledge this limitation and will discuss it more explicitly in the final version, including the cross-domain generalization evidence. We also agree that integrating the objective into full pre-training is a promising future direction.
>
> > **W1-2: Extra forward passes to safeguard parallelization.**
>
> We would like to clarify that SPEED itself does not require extra forward passes at inference: using the distilled model alone (SPEED-LLaDA / SPEED-Dream) follows the same semi-autoregressive decoding pipeline as the base model. The JSD-based decoding is optional and is mainly used as a robustness layer when decoding becomes more aggressive or tasks are harder. Importantly, in our implementation, computing JSD does not add additional model evaluations: we reuse logits that are already produced at the first step of decoding adjacent blocks, and then compute the JSD value directly from these cached logits. The main overhead of enabling JSD is indirect: the fast/slow partition can introduce a small number of extra decoding groups within a block (i.e., a few more model forward passes), trading a bit of speed for improved robustness. We will clarify this in the implementation details and explicitly discuss the overhead/benefit trade-off in the revision.
>
> > **Q1: Generalization capability of SPEED.**
>
> Thanks for raising this important question. We expect the speedup to generalize across domains because SPEED does not rely on task-specific labels or reward signals. Concretely, the training objective aims to make correct tokens become confidently resolvable earlier rather than memorizing patterns from any particular benchmark. To test cross-domain behavior beyond math/code, we additionally evaluate on CNN/DailyMail [1] (news summarization) and TruthfulQA [2] (truthfulness-focused QA designed to trigger misconceptions), both of which differ substantially from existing code and math reasoning tasks in the main paper.
>
> | Method         | CNN/DailyMail | TruthfulQA  |
> | :------------- | :-----------: | :---------: |
> | Original LLaDA |  19.7 (1.0×)  | 41.8 (1.0×) |
> | SPEED-LLaDA    |  20.8 (3.4×)  | 40.6 (7.1×) |
>
> We evaluate SPEED-LLaDA with max generation length 256, block size 32, and the same entropy-threshold decoding as in the main paper (threshold $0.5$). The results show that SPEED preserves task performance while providing substantial acceleration in both domains, with smaller gains on CNN/DailyMail likely reflecting lower inherent parallelizability for summarization-style generation. Importantly, SPEED is not expected to be slower than sequential one-token decoding in the worst case: if few tokens satisfy the threshold at a step, the decoding rule naturally falls back to decoding only the single most confident token (i.e., the greedy behavior), so the procedure does not incur more steps than the sequential baseline under the same implementation.
>
> [1] "Teaching machines to read and comprehend." Advances in neural information processing systems, 2015
>
> [2] "Truthfulqa: Measuring how models mimic human falsehoods." In Proceedings of the 60th annual meeting of the association for computational linguistics, 2022.

---

> > ### Author Rebuttal · Reviewer_UQU5 · 2026-04-02
> >
> > I've read the response. Although my concerns are all addressed, after reading other reviewers' opinion, I think the current draft does not yet fit for the next level of confidence in acceptance. I'll keep my rating.

---

> > > ### Author Response · Authors · 2026-04-04
> > >
> > > We sincerely thank you for your feedback and for confirming that our response has adequately addressed your original concerns. We will incorporate the clarifications and additional evaluations into the revision to strengthen the paper’s overall clarity and justification. We also welcome any further feedback that could help improve the final version.

---

### Decision · Program_Chairs · 2026-04-30

**Decision:**

Accept (regular)

**Comment:**

The reviewers generally agreed that the paper addresses an important problem in diffusion LLM decoding, and the overall discussion leans toward acceptance. They found the token-grouping perspective insightful and viewed the combination of sharpened teacher distillation and Slow–Fast decoding as an effective way to improve the speed–quality tradeoff. The empirical results were broadly seen as strong, with substantial speedups and limited quality loss across the main benchmarks. While some concerns remained about the extra post-training cost, the reliance on teacher-generated data, and the relatively modest added benefit of the JSD component, these issues were largely clarified in the rebuttal and did not substantially weaken the overall case for the paper.